# 🤖 Spatial CAPTCHA: Generatively Benchmarking Spatial Reasoning for Human-Machine Differentiation

**Arina Kharlamova**[*,1]**, Bowei He**[*,1,2]**, Chen Ma**[3]**, Xue Liu**[1,2]

[1] MBZUAI, [2] McGill University, [3] City University of Hong Kong

{Arina.Kharlamova, Bowei.He}@mbzuai.ac.ae

## Abstract

Online services rely on CAPTCHAs as a first line of defense against automated abuse, yet recent advances in multi-modal large language models (MLLMs) have eroded the effectiveness of conventional designs that focus on text recognition or 2D image understanding. To address this challenge, we present **Spatial CAPTCHA**, a novel human-verification framework that leverages fundamental differences in spatial reasoning between humans and MLLMs. Unlike existing CAPTCHAs which rely on low-level perception tasks that are vulnerable to modern AI, Spatial CAPTCHA generates dynamic questions requiring geometric reasoning, perspective-taking, occlusion handling, and mental rotation. These skills are intuitive for humans but difficult for state-of-the-art (SOTA) AI systems. The system employs a procedural generation pipeline with constraint-based difficulty control, automated correctness verification, and human-in-the-loop validation to ensure scalability, robustness, and adaptability. Evaluation on a corresponding benchmark, **Spatial-CAPTCHA-Bench**, demonstrates that humans vastly outperform 10 state-of-the-art MLLMs, with the best model achieving only 31.0% Pass@1 accuracy. Furthermore, we compare Spatial CAPTCHA with Google reCAPTCHA, which confirms its effectiveness as both a security mechanism and a diagnostic tool for spatial reasoning in AI.

## 1 Introduction

Modern web services face persistent threats from automated abuse, including credential stuffing, content scraping, and spam. To mitigate these risks, **CAPTCHAs** (Completely Automated Public Turing tests to tell Computers and Humans Apart) pose challenge–response tests that are easy for humans yet hard for machines, serving as a practical, first-line defense at Internet scale (Von Ahn et al., 2003). In fact, CAPTCHA technologies have been widely commercialized and deployed across all major web platforms like Google and Facebook, e-commerce services, and security infrastructures (Kumar et al., 2022). Unlike general-purpose evaluation suites such as Human's Last Exam (Phan et al., 2025) and ZeroBench (Roberts et al., 2025), CAPTCHAs must be automatically and continuously generated, remain unpredictable, and preserve a human–machine difficulty gap in the wild.

In the past decades, CAPTCHA mechanisms have evolved from text-based and image-based variants to more sophisticated protocols, including Google reCAPTCHA (BuiltWith, 2024a;b), Diff-CAPTCHA (Jiang et al., 2023a), and VideoCAPTCHA (Gurale et al., 2025). However, the rapid progress of advanced machine intelligence, especially multi-modal large language models (MLLMs) such as GPT-4o (OpenAI, a), and tool-using agents (OpenAI, 2025), have enabled computers to surpass the human capabilities in many areas, making existing CAPTCHAs not reliable anymore. Especially, CAPTCHA systems that primarily test superficial pattern recognition (e.g., object detection) are increasingly vulnerable (Deng et al., 2024); even adversarial hardening can only yields transient robustness with limited generalization (Hitaj et al., 2020).

---

[*]Equal contribution

However, in spite of achieving the great success on many language and 2D perception tasks, such MLLMs/agents still exhibit significant limitations on spatial understanding and reasoning, largely due to the scarcity of related training data and the constraints of current visual encoder designs (Wu et al., 2025a; Xu et al., 2025b). In contrast, humans possess innate 3D perceptual and spatial-reasoning capacities, which arise from genetic predispositions and are further refined by postnatal sensory–motor experience and cultural/environmental learning (Mallot & Basten, 2009). In other words, humans inherently have an internal spatial model in their minds and thus construct the 3D scenario with only a single-perspective image (Land, 2014). This motivates us to utilize this characteristic to distinguish human and machines.

To achieve this goal, we first categorize and design seven types of tasks to evaluate spatial capabilities which are easy for humans but challenging for AI (MLLMs). Then, we develop an autonomous pipeline, **Spatial CAPTCHA**, which can generate unlimited questions corresponding to each task, suitable for real-world online service In particular, we have integrated mechanisms including constraint-based difficulty control, automated correctness verification, and human-in-the-loop validation to ensure scalability, robustness, and adaptability. Further, we collect a certain number of generated instances from each task to obtain a benchmark, **Spatial-CAPTCHA-Bench**, thus evaluating the performance of different testers in an offline manner. We evaluate the human and machine performance on this benchmark and also questions from representative CAPTCHAs including Google reCAPTCHA. Experiment results demonstrate advanced MLLM's scores on our benchmark are much lower than those on Google reCAPTCHA, especially for most advanced models (e.g., 29.0 vs 55.3 for Gemini-2.5-Pro). Meanwhile, the human scores can keep consistently over 90 similar to other CAPTCHAs, which indicates our spatial CAPTCHA can effectively differentiate human and machines.

## 2 RELATED WORKS

**Bot Attacks and Defense:** Bot attacks, automated scripts or agents that mimic human interactions, abuse online services through content theft, inventory scalping, payment fraud, account takeover, or infrastructure overload, posing serious security and economic risks (Dunham & Melnick, 2008; Kumar et al., 2022). They cause direct financial loss and erode customer trust, with industry reports confirming their prevalence and costliness (Imperva, 2025). Recent attacks on human-verification systems evolve along two axes (Plesner et al., 2024): (i) powerful vision and vision–language models (Liu et al., 2023; OpenAI, a) generalize across CAPTCHA types and defeat unseen challenges (Teoh et al., 2025); (ii) behavioral simulation via generated mouse, touch, or timing patterns enables bypassing detectors (Liu et al., 2024). As a result, adversaries now combine solvers (Motoyama et al., 2010; Ye et al., 2018), behavior emulation, and adaptive strategies (Deng et al., 2024), motivating CAPTCHAs that probe reasoning modalities where humans still retain advantage (Hitaj et al., 2020). Therefore, we introduce Spatial CAPTCHAs that require relational and spatial reasoning, which is robust against current multimodal solvers and behavioral mimics.

**MLLMs and Agents:** Recent years have seen rapid progress in multimodal large language models (MLLMs), spanning open-source efforts (e.g., the BLIP family (Li et al., 2022; 2023; Dai et al., 2023), LLaVA series (Liu et al., 2023; Lin et al., 2024), and Qwen-VL (Wang et al., 2024b; Bai et al., 2025)) and proprietary systems (e.g., Claude 4 (Anthropic), Gemini 2.5 (Google DeepMind, a;b), GPT-4o (OpenAI, a), and GPT-4o mini (OpenAI, b)). Their canonical modular design—where a vision encoder extracts features that are aligned via a projection layer before being fed into an LLM—enables seamless multimodal understanding and generation. Powered by large-scale pretraining, MLLMs excel on tasks such as visual question answering, OCR, and reasoning over diagrams or videos (Liu et al., 2023), and they underpin practical agents like GUI agents (Wang et al., 2025; OpenAI, 2025) that interpret screen content to execute actions. Despite these advances, current models remain limited in spatial reasoning (Xu et al., 2025b): unlike humans, who can infer 3D structures and dynamics from partial observations, MLLMs often fail on tasks requiring geometric consistency, physical intuition, or embodied perspective-taking. This gap motivates CAPTCHAs that exploit machine weaknesses in spatial reasoning.

## 3 THEORETICAL BASIS OF SPATIAL CAPTCHA

The Spatial CAPTCHA paradigm is grounded in well-studied human cognitive abilities rather than arbitrary puzzle design. Human spatial cognition is characterized by several fundamental abilities

(Porat & Ceobanu, 2024; Freksa et al., 2017), including **(I)** spatial perception and reference system, **(II)** spatial orientation and perspective-taking, **(III)** mental objects rotation and **(IV)** spatial visualization involving multiple transformations. These categories have been identified in psychometric taxonomies (Bar-Hen-Schweiger & Henik, 2024; Carroll, 1993; Knauff, 2006) and operationalized in classic instruments (Shepard & Metzler, 1971; Hegarty & Waller, 2004; Duffy et al., 2024).

Spatial CAPTCHA formalizes each spatial ability as a distinct task category, where the solution is anchored in a mathematically well-defined invariant. These invariants include but not limited to topological relations to coordinate transformations (Stevens et al., 2012; Cohn & Renz, 2008; Egenhofer & Franzosa, 1991), rotational equivalence in two and three dimensions (Shepard & Metzler, 1971; Cohen et al., 2018; Cohen & Welling, 2016), and the composition of Euclidean motions (Murray et al., 1994; Lynch & Park, 2017; Blanco, 2010). Their concrete parameterization including covering input variables, rendering constraints, and answer definition is specified in task manifests, described in detail in §4.2. Concretely, an instance is produced by sampling from a parametric family $x \sim \mathcal{G}(\theta)$ with $\theta \sim P_\Theta$, where the associated query $f(x)$ explicitly targets the intended invariant. This design, combined with constraint-based modulation of visual cues, ensures that task success depends on genuine spatial reasoning rather than incidental lexical patterns or surface textures.

Our contribution is a theory-first framework that maps cognitive constructs onto verifiable invariants, yielding a compositional ensemble of task classes. For completeness, Appendix A presents all four ability categories with representative task instances.

## 4 SYSTEM FRAMEWORK OF SPATIAL CAPTCHA

### 4.1 INVARIANT-SPECIFIED TASK MANIFESTS AND GROUND-TRUTH CERTIFICATION

Before introducing the detailed procedural mechanics of generation and rendering, we first begin from the declarative level by formally specifying the structure of invariant-specified task manifests and the certification rules that guarantee their validity as ground truth: namely, *what* can be generated and *how* it is certified.

**What a manifest is (concrete representation)** A *task manifest* is a machine–checkable specification that binds a cognitive invariant to a family of renderable items with controlled variability and difficulty. In practice, manifests are canonical JSON objects validated against a JSON Schema; the schema, validators, and CLI tooling which are described in §4.2. Formally, a manifest is a tuple

$$\mathcal{M} = \langle id, \ I, \ (\Theta, P_\Theta), \ \mathcal{T}, \ \mathcal{G}, \ \Gamma, \ \mathcal{V}, \ \mathcal{R} \rangle,$$

For clarity and reproducibility, we now state the role and type of every field in $\mathcal{M}$: **(I)** $id \in \mathsf{ID}$ is the manifest identifier (name, type, version) ensures provenance and reproducibility. **(II)** $I \in \mathsf{Inv}$ is the targeted invariant which captures the semantics of the class of tasks and everything else merely serves this check (e.g., left/right allocentricity, rotational congruence, topological adjacency). **(III)** $(\Theta, P_\Theta)$, where $\Theta = \{\theta_i\}, P_\Theta \in \Delta(\Theta)$ is the *concrete parameterization* of content variables with a sampling prior $P_\Theta$ (counts, angles, poses, occlusions, candidate set size, distractor types) that defines the input space, where each parameter has a well-typed domain (ranges, enumerations, or stochastic permutations). **(IV)** $\mathcal{T} : \Theta \times \mathsf{S} \to (\Sigma^*, \ \mathrm{Ans}, \ a^\star)$ is *the task* function that instantiates the question, candidate set, and correct answer, binding scene semantics to a solvable problem. **(V)** $\mathcal{G} : \Theta \to \mathsf{S}$ is the *scene function*, a pseudo-random generator that from $\Theta$ constructs a candidate world model, produces derived outputs (e.g., answer key), and encodes geometry in a coordinate-based structure. **(VI)** $\Gamma = (\Gamma_{\text{false}}, \Gamma_{\text{slots}})$ adds distractor mechanisms, producing false answers or slot fillers from the scene so that multi-option tasks remain nontrivial. **(VII)** $\mathcal{V} : \mathsf{S} \to \{0, 1\}$ is the *validator suite*, rejecting invalid scenes (e.g., intersecting objects, insufficient margins, lack of uniqueness) and ensuring well-posedness. **(VIII)** $\mathcal{R} : \Theta \times \mathsf{S} \to \mathsf{X}$ is the *renderer*, projecting the validated scene into images or panels with fixed style settings.

**Minimal guarantees** All components operate in geometric space: $\mathcal{G}$ constructs scenes by rigid motions (placing objects under explicit spatial relations), $\Gamma$ produces near–miss candidates via controlled spatial perturbations, and $\mathcal{V}$ verifies invariants (non-intersection, adjacency, and separation margins encoded by $\Gamma$). Consequently:

- *soundness*: the label $y$ is computed from the scene $S$ and is independent of rendering;

- *uniqueness under margins*: if $\Gamma(S) = 1$, exactly one candidate in the set returned by $\mathcal{T}$ satisfies the predicate family tied to $I$;

- *validity and human legibility*: visibility/contrast and margin checks reject ambiguous or visually marginal items; and

- *spatial necessity*: success requires the intended spatial reasoning, since distractors differ only in prohibited relations while superficial appearance alone cannot satisfy the certified invariants.

**Difficulty and variability by design** Difficulty is controlled at the level of the manifest rather than left to rendering accidents. The content space $\Theta$ exposes interpretable knobs listed in details in Appendix(B.1. We define a monotone difficulty map

$$d(\theta) = w^\top \phi(\theta),$$

with features $\phi(\theta)$ drawn from these knobs; $w$ is fitted to pilot human response times via isotonic/quantile regression. Sampling uses stratified priors $P_\Theta$ over target bins (easy/medium/hard) with rejection against $\mathcal{V}$ to ensure admissibility. This keeps items *human–simple* (defined in §6.1): ambiguity is excluded by separation margins, legibility guards (visibility/contrast), and symmetry screens, while reasoning load is set by $d(\theta)$, not by clutter or texture. The detailed procedure described in Appendix B.2.

## 4.2 Instance Synthesis Pipeline

The instance synthesis pipeline turns a high-level manifest specification $\mathcal{M}$ into deliverable items with consistent difficulty control and auditability. We factor the pipeline into three macro-stages: **(I)** Scene Metadata Random Generation, **(II)** Procedural Generation, and **(III)** Task Generation, across which the internal steps are *distributed*: Sampling occurs in Stage 1; Scene Construction, Distractor Synthesis, and Validation occur in Stage 2; Rendering, Prompt-and-Answer Construction, and Assembly occur in Stage 3. This structure isolates responsibilities, lets task families evolve without cross-coupling, and preserves reproducibility across engines and environments.

**Operational Setup** *Inputs:* a valid manifest reference; a random seed; access to a scene generator, a distractor mechanism, a validator suite, and a renderer. *Outputs:* a packaged instance containing rendered panels, a prompt, an answer set with one correct choice, and metadata sufficient for re-execution.

**Scene Metadata Random Generation** Input variables $\theta \sim P_\Theta$ are drawn according to the manifest. These knobs encode the semantic degrees of freedom of the task (e.g., counts, layout, base geometry) while stratified priors control difficulty bins.

### 4.2.1 Procedural Generation

**(1) Scene generation** The sampled input $\theta$ is passed to the scene function $\mathcal{G}$, which constructs a candidate world model $S$ in geometric space. This stage employs constrained procedural generation: input variables define the admissible complexity of the scene and, later in rendering, its visual appearance, while validity functions impose additional constraints that guarantee readability and distinguishability between correct and distractor answers.

**(2) Distractor synthesis** Given $S$, distractor mechanisms $\Gamma = (\Gamma_{\text{false}}, \Gamma_{\text{slots}})$ generate near–miss alternatives. These are constrained perturbations of the base scene that yield plausible but incorrect answer candidates (e.g., wrong viewpoint, mismatched rotation, or inconsistent projection).

**(3) Validation** The validator suite $\mathcal{V}$ certifies the instance. Validators reject degenerate or ambiguous cases by enforcing invariants such as

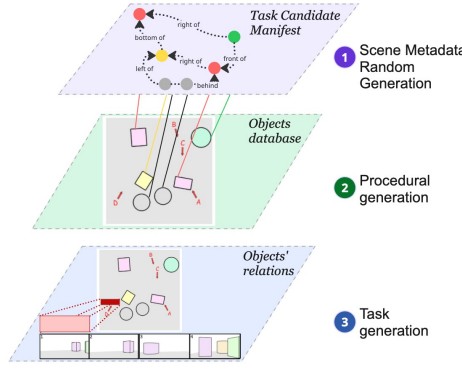

Figure 1: End-to-end synthesis pipeline. Input variables are sampled from the manifest.

Table 1: Comparison with CAPTCHA suites and spatial reasoning datasets. Columns: Modality/Eval (dominant input signal and scoring protocol), Procedural Gen (programmatic instance synthesis), Offline Supervision (public static ⟨input,target⟩ pairs suitable for supervised fine-tuning), Complexity Metric (explicit difficulty/robustness measure), Human pass (%), Best MLLM/Agent (%; strongest pass@1 reported by the source under its primary protocol), and Open Data/Code. Percentages are absolute; "–" denotes not reported.

| Benchmark / System | Modality / Eval | Procedural Gen | Offline Supervision | Complexity Metric | Human pass (%) | Best MLLM/ Agent (%) | Open Data/Code |
|---|---|---|---|---|---|---|---|
| *CAPTCHA Suites* | | | | | | | |
| (Luo et al., 2025) | Image+Text / Agentic | ✗ | ✗ | ✓ | 93.3 | 40.0 | ✓ |
| (Wu et al., 2025b) | Image+Text / Offline | ✗ | ✓ | ✗ | 98.0 | 99.5 | ✓ |
| (Ding et al., 2025) | Image / Offline | ✓ | ✓ | ✗ | 86.95 | 0.0 | ✗ |
| (Jiang et al., 2023b) | Image / Offline | ✓ | ✗ | ✗ | – | – | ✗ |
| (Chandra et al., 2025) | Audio+Video / Offline | ✓ | ✗ | ✗ | 92.8 | 5.2 | ✗ |
| *Spatial datasets* | | | | | | | |
| (Ma et al., 2025) | Image+Text / Offline | ✓ | ✓ | ✓ | 95.7 | 52.0 | ✓ |
| (Wang et al., 2024a) | Image+Text / Offline | ✓ | ✓ | ✗ | – | 67.1 | ✓ |
| (Du et al., 2024) | Image+Text / Offline | ✓ | ✓ | ✗ | 90.3 | 49.1 | ✓ |
| (Stogiannidis et al., 2025) | Image+Text / Offline | ✓ | ✗ | ✗ | – | 48.8 | ✗ |
| (Comsa & Narayanan, 2023) | Text / Offline | ✗ | ✓ | ✗ | 93.5 | 88.3 | ✓ |
| (Rodionov et al., 2025) | Text / Offline | ✓ | ✗ | ✓ | – | 85.0 | ✗ |
| *Spatial-CAPTCHA (Ours)* | | | | | | | |
| Spatial-CAPTCHA-Bench | Image+Text / Offline | ✓ | ✓ | ✓ | 99.8 | 31.0 | ✓ |

non–intersection, sufficient angular or depth margins, uniqueness of the correct answer, and visibility/contrast checks. Only scenes with $\mathcal{V}(S) = 1$ are admitted.

### 4.2.2 TASK GENERATION

**(1) Rendering** The validated scene is mapped to images via $\mathcal{R} : \Theta \times S \rightarrow \mathsf{X}$, producing one or more rendered panels. Rendering is label–inert: it affects visual style but not the computed answer. In practice, this stage may call external engines such as Blender for high–fidelity 3D output or VTK for lightweight geometric visualization, but the pipeline itself remains agnostic to the rendering backend.

**(2) Prompt and answer construction.** Input variables $\theta$ and outputs of $\mathcal{G}$ are bound into a task template $\mathcal{T}$, producing the natural–language prompt, the candidate set, and the correctness marker. Distractor variants generated in step (3) populate the answer slots.

**(3) Assembly** All components (such as rendered images, task prompt, answer variants, and correctness label) are packaged into a single CAPTCHA instance. Each instance is both service–ready (deliverable to end users) and dataset–ready (loggable for evaluation).

This design enforces two layers of constraints. First, input variables $\Theta$ control task complexity and visual load through interpretable knobs. Second, $\mathcal{V}$ enforces admissibility constraints to guarantee legibility, uniqueness, and spatial necessity. Because the pipeline is defined in terms of declarative manifests, it remains agnostic to specific rendering engines or scene implementations. This abstraction enables extensibility across task families and supports runtime instance generation personalized to user history and trust scores, without compromising the certification guarantees.

## 5 SPATIAL-CAPTCHA-BENCH: DATASET

Spatial-CAPTCHA-Bench is the first benchmark instantiated from the Spatial-CAPTCHA framework. It comprises $K=4$ spatial-ability categories (reference systems; orientation/perspective-taking; mental rotation; multi-step spatial visualization), each stratified into $D=3$ difficulty bins (easy/medium/hard). Across $T$ task formulations (currently $T=7$; extensible; e.g., Unfolded, Sun Direction, Revolution, Pyramid, Polyomino, Full Views), the dataset contains $N_{\text{inst}}=1050$ instances with per-formulation counts $(150, \ldots, 150)$ and per-bin counts $(N^{\text{E}}, N^{\text{M}}, N^{\text{H}})=(500, 300, 250)$. Per-category counts are $(N_1, \ldots, N_4)$ with $\sum_{i=1}^{4} N_i = 1050$. Table 1 contrasts Spatial-CAPTCHA-Bench with both

Table 2: Results on Spatial-CAPTCHA-Bench and reCAPTCHA-Bench. The left 3 columns reports aggregate metrics; the middle 4 columns reports pass@1 by specific abilities: spatial perception (reference systems), spatial orientation (perspective-taking) , mental object rotation, and multi-step spatial visualization. They are abbreviated as SP, SO, MOR, and SV in the table, respectively. The right 2 columns reports pass@1 and pass@$k$ on reCAPTCHA-Bench. Higher score indicates better performance.

| | | Spatial-CAPTCHA-Bench | | | | | | | reCAPTCHA-Bench | |
| | | Overall Metrics | | | Per-Ability Pass@1 | | | | Overall Metrics | |
| Methods | Rank | pass@1 | pass@$k$ | $k$-of-$k$ | SP | SO | MOR | SV | pass@1 | pass@$k$ |
|---|---|---|---|---|---|---|---|---|---|---|
| *Baseline* | | | | | | | | | | |
| Chance level (Random) | – | 21.4 | 51.1 | 1.1 | 16.7 | 25.0 | 16.7 | 25.0 | 0.2 | 0.6 |
| Human Level (Simple) | – | 89.5 | – | – | 96.7 | 95.6 | 89.6 | 83.3 | 86.4 | – |
| *Proprietary Models* | | | | | | | | | | |
| gpt-5.1 | 1 | 36.1 | 61.6 | 18.2 | 68.3 | 35.6 | 42.2 | 27.5 | – | – |
| gemini-3-pro-image-preview | 2 | 32.7 | 58.4 | 8.9 | 50.0 | 40.3 | 32.1 | 25.8 | – | – |
| o4-mini | 3 | 31.0 | 56.0 | 10.3 | 60.0 | 35.7 | 31.6 | 25.3 | 36.7 | 54.0 |
| gemini-2.5-pro | 4 | 29.0 | 48.4 | 9.9 | 44.0 | 31.7 | 30.7 | 23.7 | 55.3 | 58.7 |
| chatgpt-4o-latest | 5 | 26.1 | 38.0 | 17.7 | 44.0 | 23.3 | 27.1 | 27.3 | 52.7 | 57.3 |
| gemini-2.5-flash | 8 | 21.6 | 44.6 | 6.0 | 16.7 | 25.0 | 16.7 | 25.7 | 31.3 | 40.0 |
| claude-sonnet-4 | 10 | 21.4 | 30.8 | 11.0 | 24.0 | 21.7 | 18.0 | 26.3 | 10.7 | 15.3 |
| claude-opus-4 | 12 | 7.1 | 13.0 | 2.1 | 4.7 | 5.3 | 2.0 | 16.7 | 6.0 | 7.3 |
| *Open-weight Models* | | | | | | | | | | |
| qwen2.5-vl-72b-instruct | 6 | 24.0 | 31.0 | 16.2 | 34.7 | 23.0 | 21.6 | 28.7 | 4.0 | 6.0 |
| phi-4-multimodal-instruct | 7 | 22.7 | 32.9 | 11.4 | 19.3 | 27.7 | 20.2 | 21.3 | 2.7 | 2.7 |
| llama-4-maverick | 9 | 21.5 | 29.9 | 12.7 | 13.3 | 28.7 | 14.7 | 24.7 | 2.7 | 3.3 |
| mistral-medium-3 | 11 | 20.2 | 43.5 | 4.7 | 14.0 | 27.7 | 13.6 | 22.7 | 6.7 | 12.0 |

CAPTCHA suites and spatial reasoning datasets. Beyond comparative positioning, the scale and coverage of Spatial-CAPTCHA-Bench open qualitatively new research directions. The dynamic extensibility of the dataset also enables forward-looking experimentation: researchers can introduce new spatial invariants, difficulty progressions, and distractor families without breaking comparability.

# 6 EXPERIMENTS

## 6.1 EXPERIMENTAL SETUP

We evaluate model and human performance on Spatial-CAPTCHA-Bench(see §5). For human evaluation, we additionally construct a Spatial-CAPTCHA-Bench (Tiny) subset of 70 items, stratified by task category and difficulty level. We also conduct experiments on reCAPTCHA-Bench, a dataset with 150 samples collected from Google reCAPTCHA service (Plesner et al., 2024; BuiltWith, 2024b).

**Models Evaluated** We assess a diverse pool of state-of-the-art Large Language and Vision–Language Models, spanning both proprietary (e.g., GPT-4o (OpenAI, a), Claude Sonnet 4 (Anthropic), Gemini 2.5 Pro (Google DeepMind, b)) and open-source architectures (e.g., Llama, Mistral). The complete list is provided in Table 2. All models are evaluated zero-shot without fine-tuning or chain-of-thought augmentation.

**Evaluation Metrics** We adopt a multi-faceted evaluation protocol designed to capture both task-level correctness and cognitively grounded failure patterns. Evaluation metrics are grouped by intent into accuracy, human upper bound, and ability-specific diagnostics. A detailed Appendix D summarises the full set of metrics used throughout this study, along with their scope and interpretive roles. Specifically, $k$ is set as 3.

**Evaluation Process.** Each model is evaluated independently per task instance. Prompts are held fixed across all runs and models; no instance-level tuning is permitted. Human annotators (N=60) were instructed to solve each Tiny instance as fast as possible, simulating the "human-simple" requirement. All codes and prompts are provided in the Github repository to ensure reproducibility.

## 6.2 BASELINES

**Chance-Level (Random)** A trivial baseline selects uniformly at random from the candidate answer set. This reflects a calibrated floor of performance for each task formulation. Due to class imbalance and distractor synthesis, random accuracy varies slightly across task types, but remains within $21.4\%$ across the benchmark.

**Human-Level (Simple)** To approximate a soft upper bound, we report a **Human-Simple Pass Rate** on a 70-instance subset (Spatial-CAPTCHA-Bench (Tiny)), annotated by $N = 60$ human raters under a 30-second time constraint per item. An item is marked as "passed" if at least two annotators select the correct answer. This simulates low-friction human reasoning under minimal supervision.

## 6.3 EXPERIMENTAL RESULTS AND ANALYSIS

**Comparison with reCAPTCHA-Bench** From Table 2, relative to reCAPTCHA-Bench, it can be observed that the scores on reCAPTCHA-Bench are much higher than that on our Spatial-CAPTCHA-Bench for advanced MLLMs (e.g., 29.0 vs. 55.3 for Gemini-2.5-Pro). As for human evaluation, the score on Spatical-CAPTCHA-Bench (tiny) is even a little bit higher than that on reCAPTCHA-Bench. This demonstrates our Spatial CAPTCHA can indeed better differentiates human from machines by identifying larger human-model gap. This also proves the superiority and the potential for large-scale commercial use of our designed Spatial CAPTCHA.

**Efficiency and accuracy are only weakly coupled.** As shown in Figure 2c, latency spans two orders of magnitude across systems, yet slower models are not more accurate: Gemini-2.5-Pro exhibits the largest median response time (95.4s, IQR [17.6, 160.5]) without a commensurate accuracy advantage, while Gemini-2.5-Flash answers in near real time (1.8s, IQR [1.6, 2.2]) with only modest losses. High-latency models such as phi-4 and qwen2.5-vl-72b similarly fail to convert time into accuracy, suggesting inefficiency rather than deeper reasoning. Moreover, the variance profiles differ sharply: some models (e.g., Gemini-2.5-Pro) fluctuate by over an order of magnitude, whereas others (e.g., o4-mini, Gemini-Flash) remain stable, indicating that latency is more diagnostic of system implementation and routing overhead than of spatial reasoning capability.

**Task characteristics are the cause of systematic differences observed between humans and models.** Radar plot Figure 3b show that accuracy peaks on SUN DIRECTION and PYRAMID, where reconstruction based on sequential signals is sufficient, but collapses on UNFOLDED and AGENT SIGHT, which require enforcing adjacency constraints or integrating occluded multi-view geometry. As highlighted in Figure 2a, humans display near-reflex latencies on SUN DIRECTION (median

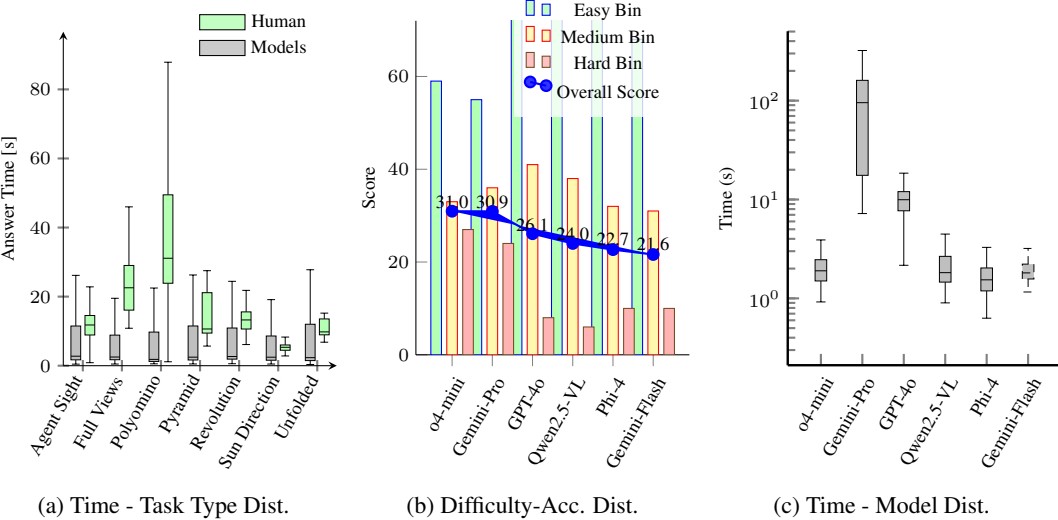

(a) Time - Task Type Dist.    (b) Difficulty-Acc. Dist.    (c) Time - Model Dist.

Figure 2: Distributions of response times and accuracies across task types, difficulty levels, and models.

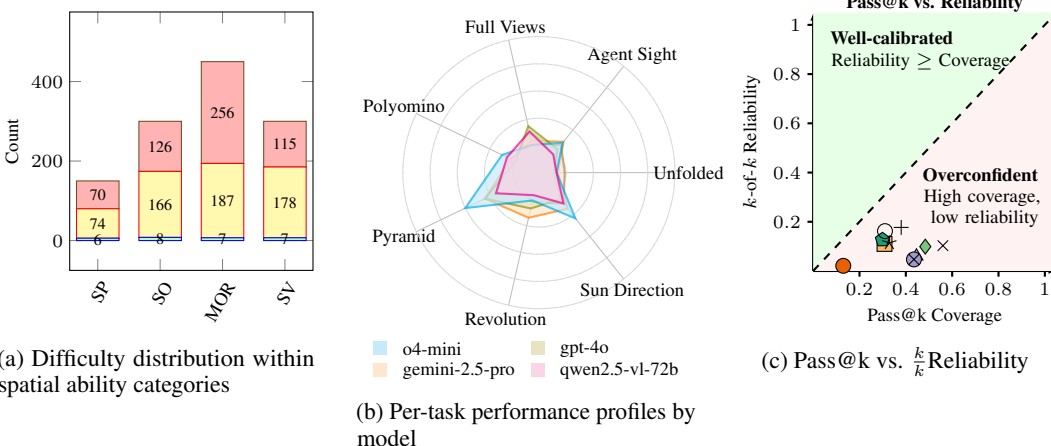

(a) Difficulty distribution within spatial ability categories

(b) Per-task performance profiles by model

(c) Pass@k vs. $\frac{k}{k}$ Reliability

Figure 3: Overview of task difficulty, model profiles, and reliability. Colours in (b,c) mark top-5 models from Table 2, where shown results are also consistent with Figure 2.

2.1s; IQR $[1.3, 2.9]$), consistent with embodied heuristics (e.g., shadow–light vector decoding), whereas models show no analogous latency drop, which is evidence of missing perceptual grounding. The UNFOLDED family is particularly diagnostic: models answer quickly yet fail often, a pattern consistent with template-based shortcuts that ignore global compatibility. At the level of cognitive class, performance is higher for spatial perception and reference-frame alignment ($27.5\% \pm 16.6$ pp) than for multi-step visualisation ($24.2\% \pm 5.7$ pp), while human accuracy is comparatively stable across abilities (within $\pm 6.7$ pp). This consistency, contrasted with the variability observed in models, implies that the system is more sensitive to the depth of transformations rather than the mere complexity of the imagery.

**Calibration is uniformly poor.** Figure 3c illustrates $k/k$ reliability against pass@$k$ coverage (with $k$=3) places every model beneath the identity line: confident sets under-represent the truth. `GPT-o4-mini`, for instance, achieves high coverage ($56.0$) but low reliability ($10.3$), typifying overconfidence; `claude-opus-4` is more conservative (coverage $13.0$; reliability $2.1$) yet still uninformative as none approach parity.

This consistent lack of calibration compounds the identified performance shortcomings, as models not only overestimate their certainty in individual predictions but also struggle to retain accuracy when faced with escalating task complexity. Difficulty stratification (illustrated in Figure 3a) confirms that our bins capture real complexity gradients rather than noise. Performance curves in Figure 2b indicate that from Easy to Hard, models' pass@1 drops steeply ($61.4\% \pm 48.7$ pp to $12.4\% \pm 33.0$ pp; Cohen's $h$=1.08), while humans decline gradually (slope $\approx 3.0$ pp). The combination of steep model slope and shallow human slope implies that what is hard here is not low-level vision but *compositional constraint satisfaction*: chaining local signals under global geometric rules. This aligns with the per-task anomalies above and with the observation that added latency seldom recovers correctness.

Taken together, Spatial CAPTCHA separates humans from MLLMs by diagnosing structural failures in invariant preservation, embodied perception, and calibration. This makes it both an effective discriminator and a diagnostic lens into the unresolved challenge of uncertainty-aware, constraint-preserving spatial reasoning.

# 7 CONCLUSIONS AND FUTURE WORKS

In this work, we introduced Spatial CAPTCHA, a generative framework for benchmarking and deploying spatial reasoning challenges as a new form of human–machine differentiation. By systematically designing seven categories of tasks targeting spatial understanding and reasoning, we demonstrated that our pipeline can continuously generate scalable, verifiable, and difficulty-controlled instances. Extensive evaluations revealed a persistent human–machine performance gap: while humans consistently achieved nearly 100% accuracy, state-of-the-art multimodal LLMs exhibited significant

performance drops, confirming the practicality of our approach. Moreover, the introduction of Spatial-CAPTCHA-Bench provides a reproducible offline benchmark for standardized evaluation of both human and machine capabilities. In the future, we plan to design GUI-interactive spatial reasoning challenges, requiring users to manipulate or align objects rather than simply provide answers, thereby enriching the human–machine differentiation space. Besides, extending the CAPTCHA to temporal-spatial challenges (e.g., reasoning across video sequences or dynamic object interactions) could further enhance robustness against automated solvers.

## REPRODUCIBILITY STATEMENT

For implementation details, please refer to Appendix H. The complete codebase and generation scripts required to reproduce our study are available through the `https://github.com/Doldrums/spatial_captcha`. The repository includes benchmark construction tools, evaluation pipelines, and configuration files with fixed random seeds to ensure deterministic regeneration of all benchmark instances. Detailed instructions are provided in the main text and appendices for environment setup, model evaluation with fixed zero-shot prompts, and difficulty calibration procedures. We also document the human evaluation protocol, ensuring that both machine and human baselines can be reliably reproduced. We release both the full dataset and the Tiny subset used for human evaluation on Hugging Face at `https://huggingface.co/datasets/amoriodi/Spatial-CAPTCHA-bench`.

## ETHICS STATEMENT

This study involved human participants to evaluate Spatial-CAPTCHA-Bench. Participation was entirely voluntary, and no compensation or incentives were tied to outcomes. No personal or identifying information was collected; participants could optionally provide arbitrary display names solely for leaderboard purposes. All responses were used only in aggregate analyses, and no individual-level data are reported. The study was conducted in accordance with institutional ethical guidelines, with procedures designed to minimize any potential risks to participants. The tasks involved solving spatial reasoning challenges, which posed no foreseeable risks beyond those encountered in everyday computer use. All materials, instructions, and protocols are transparently documented to ensure responsible and reproducible human evaluation.

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

# A    TASK CLASSES BY SPATIAL ABILITIES

## A.1    SPATIAL PERCEPTION AND REFERENCE SYSTEM ABILITY

Spatial perception refers to the ability to judge the arrangement and orientation of objects relative to one's frame of reference Xu et al. (2025a); Burgess (2006b). In spatial-cognition taxonomies it is treated as a core sub-ability of visuospatial reasoning. Tasks targeting this ability require the solver to detect how objects align or orient in a scene under a fixed coordinate system. Crucially, problems may be posed in egocentric (observer-centered) or allocentric (world-centered) coordinates Burgess (2006a). Such questions hinge on maintaining a consistent reference frame (e.g. a vertical axis) across views.

The key to solving spatial-perception tasks is identifying invariant geometric relations that survive rigid transformations. In particular, collinearity and parallelism are preserved under translation and rotation. For instance, points that lie on a straight line in one view remain collinear in any rotated or translated view, and any pair of parallel lines stays parallel after rotation or scaling. Humans naturally excel at judging basic alignments and reference-relationships, but this skill is difficult for algorithms lacking explicit frame-of-reference reasoning Sun & Wang (2010). By contrast, models often fail when surface textures change even though the geometry is unchanged.

In Figure 4 examples illustrate that the underlying invariant (alignment, orientation, and relative positioning across different views) is explicitly targeted. Each Spatial CAPTCHA instance is

generated by sampling a rigid transformation (rotation/translation) of a base scene and asking a question anchored on the invariant relation. Solvers must therefore track the reference axis and preserving orientation, not surface appearance.

## A.2 SPATIAL ORIENTATION AND PERSPECTIVE-TAKING ABILITY

Spatial orientation and perspective–taking is the ability to compute where things are *relative to a viewpoint* and to mentally adopt alternative viewpoints without physically moving. Cognitive science distinguishes egocentric (viewer–centered) and allocentric (world–centered) encodings, with perspective–taking requiring systematic transforms between the two Hegarty & Waller (2004); Carroll (1993); Knauff (2006). Classic findings show dissociations between object rotation and perspective–taking: the latter engages navigation- and scene–based skills (updating the heading, re-anchoring axes, handling occlusions) that are only weakly predicted by mental rotation performance Hegarty & Waller (2004).

In the collage 5, the correct answer is determined by a viewer–centered predicate invariant to world-frame rotations and translations, not by object appearance.

## A.3 MENTAL OBJECTS ROTATION ABILITY

Human spatial cognition is well-suited to 2D rotation tasks. Classic studies by Shepard and Metzler Shepard & Metzler (1971) showed that when subjects decide whether two shapes are the same under rotation, their reaction time increases linearly with the angular difference between the shapes. Introspective reports confirm that people "mentally rotate" one image to align with the other. Similarly, the Vandenberg–Kuse Mental Rotations Test (MRT) presents flat images (often of 3D-based objects or letters) at various orientations, and asks participants to identify which candidates are the same shape versus mirror reflections. These findings support an analog mental-imagery process: subjects form a mental representation of the base shape and continuously rotate it until it matches a target orientation, then make a match/mismatch decision. Representative instances that isolate this ability are shown in Figure 6.

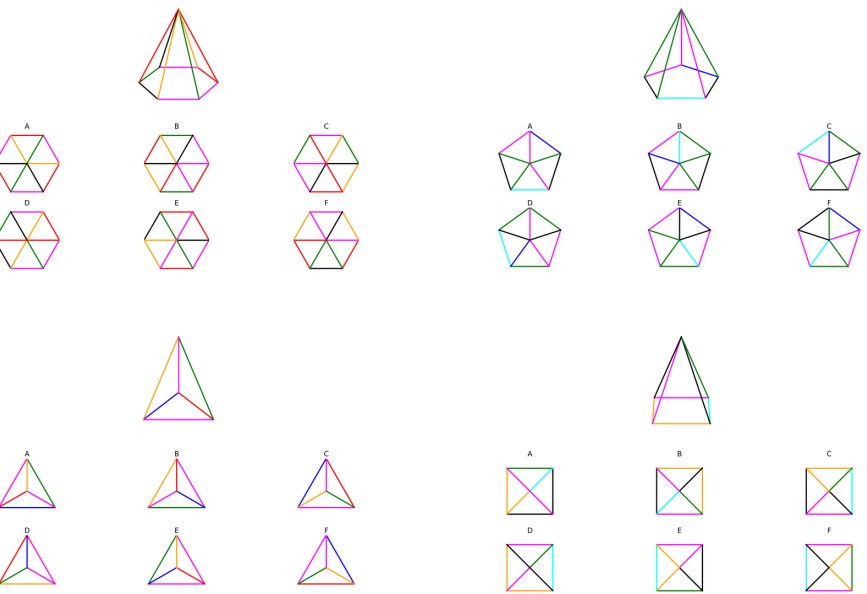

Figure 4: Illustrative examples of tasks targeting *Spatial perception and reference system ability*.

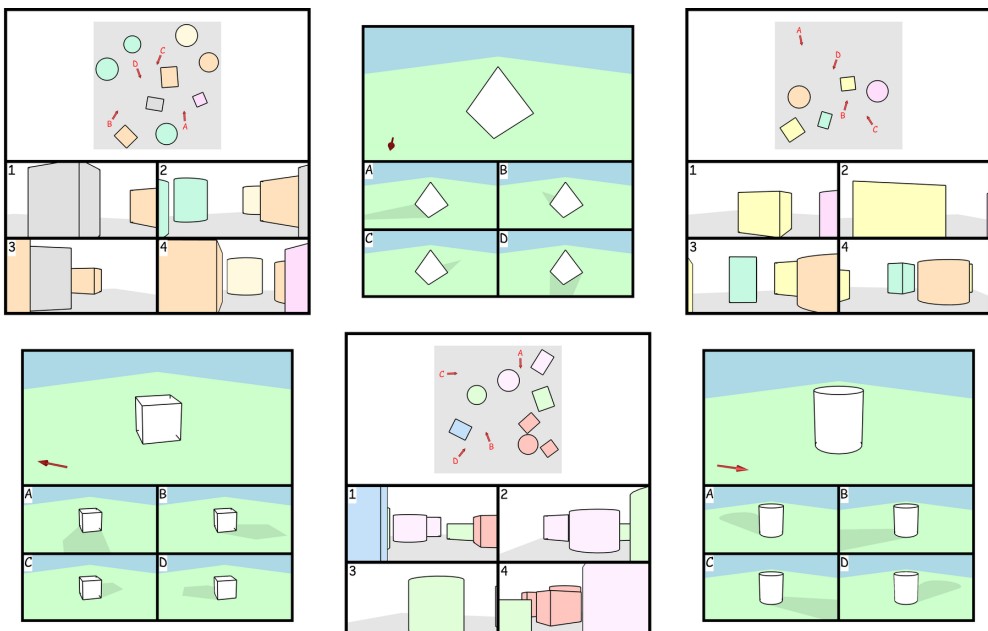

Figure 5: Examples of tasks probing *Spatial orientation and perspective-taking*. Participants must mentally adopt alternative viewpoints to determine relative positions or directions of objects. The design highlights the distinction between object-centered transformations (rotation) and observer-centered transformations (orientation shift).

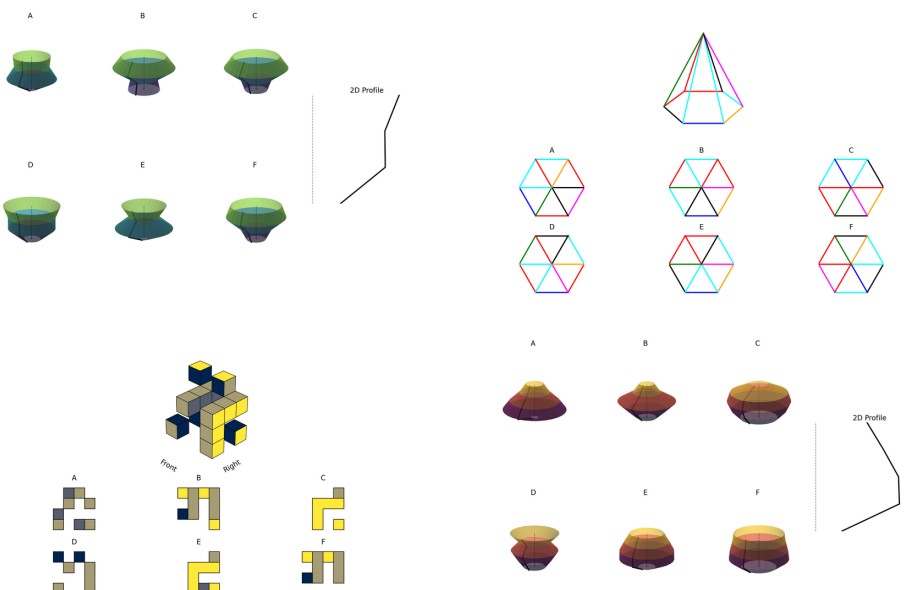

Figure 6: Examples of tasks engaging the *Mental objects rotation ability*. The settings include polyhedral matching, 3D block assemblies and abstract shape comparisons. In all cases successful performance requires mentally rotating objects to establish equivalence or detect mismatch, showing how this core capacity recurs across spatial reasoning challenges.

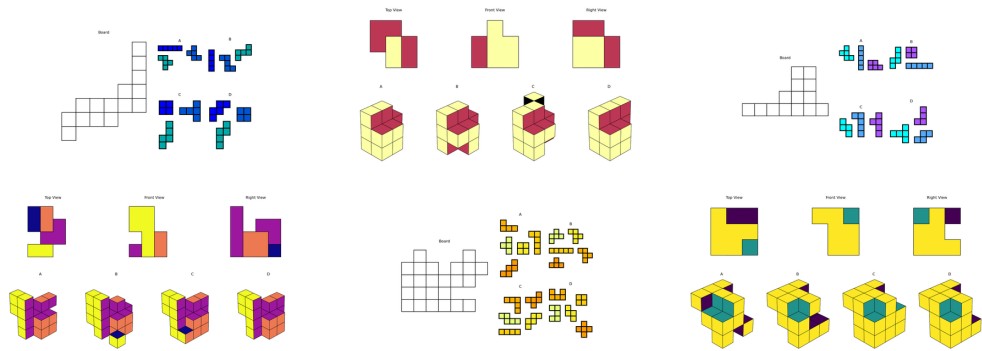

Figure 7: Examples of tasks engaging the *spatial visualization ability involving multiple transformations*.

### A.4 SPATIAL VISUALIZATION INVOLVING MULTIPLE TRANSFORMATIONS

Spatial visualization denotes the capacity to manipulate an imagined configuration through a *sequence* of operations (as rotations, reflections, translations, folds, cuts, and recombinations) while keeping track of intermediate states. In psychometrics it is treated as a factor separable from, though correlated with, mental rotation and spatial orientation Carroll (1993); Hegarty & Waller (2004). Classic instruments such as the Paper Folding Test (PFT) instantiate this ability by requiring subjects to simulate multiple fold–punch–unfold steps to predict the final pattern. Unlike single–transform problems, success depends on composing operations and maintaining a stable internal representation across steps. Representative instances that refer to this ability are shown in Fig. 7.

## B  DIFFICULTY MAP CONSTRUCTION AND CALIBRATION

### B.1 INTERPRETABLE KNOBS

For each class we vary only factors that change the spatial problem, not its appearance:

- *Perception (reference frame).* Number of objects in the scene; polygonal complexity (sides 3–8); tilt magnitude relative to gravity/horizon; minimal gaps $\delta_d$ between primitives; number of near-parallel distractors.

- *Perspective-taking.* Camera yaw/pitch/roll ranges; baseline distance to landmarks; number of landmarks; depth layers (near/mid/far) and occlusion fraction; horizon tilt; discrete viewpoint set size $m$ (candidate panels).

- *Mental rotation (2D).* Rotation angle gap $\Delta\theta$; presence/absence of mirror alternatives; vertex count/concavity of shapes; symmetry order of the base shape (to avoid trivial or ambiguous matches); number of candidates.

- *Topological relations.* Grid size/board extent; number of pieces/regions; hole count and connectivity; minimal separation between components; edit distance of distractor graphs (touching vs. strictly inside/outside).

Variability is achieved without compromising label soundness. The scene function $\mathcal{G}$ and distractor mechanisms $\Gamma$ generate semantic diversity (base shapes, layouts, camera poses, fold sequences) while remaining within the invariant $I$. Distractors are synthesized as near–misses along the same spatial axes that define $d(\theta)$ (e.g., angle gaps just above $\delta_\theta$, mirrored but non–congruent shapes, off–by–one transform sequences), so success requires the intended spatial relation rather than superficial cues.

**Reproducibility and provenance** Every instance carries a manifest identifier and seeds for $(\theta, \eta)$, enabling exact regeneration and audit. Changes to a class are diffs to $\mathcal{M}$ (versioned), not ad–hoc asset edits.

## B.2 DIFFICULTY MAP CONSTRUCTION

By elevating the manifest to the first–class abstraction, we (i) connect each item family to a precise invariant, (ii) guarantee ground–truth correctness and uniqueness independent of rendering, and (iii) unlock an effectively unbounded, auditable, and *human–simple* item bank (details in Section 5). The detailed procedure for constructing and calibrating the difficulty map, including isotonic and quantile regression fits as well as binning strategies, is provided in Appendix B.2, B.3 and B.4. An illustrative example manifest, including the field-to-symbol alignment, is presented in Appendix C.1.

## B.3 ISOTONIC AND QUANTILE REGRESSION DETAILS

The objective of this stage is to translate raw human performance statistics, primarily response times and success rates, into a calibrated difficulty signal that is both monotone and globally comparable. We formalise the mapping in two phases: (i) fitting predictive models from task parameters to human outcomes, and (ii) combining these predictions into a single latent difficulty score that can be inverted during item generation.

**Data structure** For each task family, we collect a dataset of $N$ instances. Each instance is annotated with (a) a hyperparameter vector $\mathbf{x}$ specifying the generative knobs (e.g., polygon sides, rotation angle, viewpoint set size), and (b) observed human outcomes: mean response time $t(\mathbf{x})$ on correct trials, and empirical success rate $s(\mathbf{x}) \in [0, 1]$. The goal is to characterise how variations in $\mathbf{x}$ influence human performance.

**Monotone regression of response times** Response times are positive, heavy-tailed, and expected to grow monotonically with task difficulty. We therefore apply isotonic regression to $\log t(\mathbf{x})$, fitted separately for each family. The isotonic model $\widehat{t}_f(\mathbf{x})$ learns a non-decreasing function along axes of known monotonicity (e.g., larger rotation angles, greater occlusion), optionally smoothed to avoid degenerate step functions. This yields a calibrated predictor of expected solution latency.

**Quantile modelling of success rates.** Success rates lie in $[0, 1]$ and typically exhibit heteroscedastic, non-Gaussian noise with ceiling effects on easier instances and occasional floor effects on harder ones, but not a strict bimodal pattern. To capture this variability without imposing a parametric mean–variance relationship, we fit quantile regressions for $s \mid \mathbf{x}$. The model $\widehat{s}_f(\mathbf{x})$ estimates the conditional median ($\tau{=}0.5$) as a robust central tendency and a lower quantile (e.g., $\tau{=}0.25$) to characterise fragile regions where a non-trivial fraction of participants fail despite similar knobs. Predictions are clipped to $[0, 1]$ and subsequently aligned across families via the global isotonic calibration described above.

**Unified difficulty mapping** Response time and success rate capture complementary facets of hardness: the former reflects cognitive effort given success, the latter reflects probability of failure. To fuse them, we first apply per-family rank normalisation:

$$T_f(\mathbf{x}) = \text{QuantileRank}(\log t(\mathbf{x})), \qquad E_f(\mathbf{x}) = \text{QuantileRank}(1 - s(\mathbf{x})).$$

Both $T_f$ and $E_f$ lie in $[0, 1]$, with larger values corresponding to greater difficulty. To achieve cross-family comparability, we then align these variables globally via isotonic calibration against their pooled empirical CDFs, producing $\widetilde{T}, \widetilde{E} \in [0, 1]$. The final difficulty score is defined as a convex blend

$$d(\mathbf{x}) = \alpha \, \widetilde{T}(\mathbf{x}) + (1 - \alpha) \, \widetilde{E}(\mathbf{x}), \quad \alpha \in [0, 1].$$

In practice we fixed $\alpha = 0.6$, based on preliminary trials showing that a slight emphasis on response time yields smoother difficulty distributions and better separation of adjacent levels, while still preserving discriminability from success rates. This choice is not critical but stabilises the map across heterogeneous task families.

**Inverse use in generation** During item synthesis, the difficulty map is inverted: given a target difficulty value $d^\star$ or bin, the system searches for hyperparameters $\mathbf{x}$ whose predicted difficulty $d(\mathbf{x})$ falls within the desired band. The procedure is as follows:

1. **Select a target pair.** Sample a point $(\widetilde{T}^\star, \widetilde{E}^\star)$ on the iso-difficulty line $\alpha \widetilde{T}^\star + (1 - \alpha)\widetilde{E}^\star = d^\star$, ensuring feasibility within $[0, 1]^2$.

2. **Map back to family scales.** Invert the global calibrators to obtain family-specific targets $T_f^\star, E_f^\star$, then recover approximate raw values $t^\star, s^\star$ using per-family inverse CDFs.

3. **Solve for knobs.** Search for $\mathbf{x}$ minimising

$$\lambda_t |\widehat{t}_f(\mathbf{x}) - t^\star| + \lambda_s |\widehat{s}_f(\mathbf{x}) - s^\star| + \Omega(\mathbf{x}),$$

subject to admissibility constraints (visibility margins, symmetry screens). Here $\Omega$ is a diversity regulariser encouraging coverage of the knob space.

4. **Verify.** Recompute $d(\mathbf{x})$ for the candidate $\mathbf{x}$ and accept if $d(\mathbf{x}) \in \mathcal{I}$ and prediction errors are within tolerances $(\epsilon_t, \epsilon_s)$. Otherwise, adjust the target pair along the iso-difficulty line and repeat.

This inversion procedure exploits the fitted forward models $\widehat{t}_f, \widehat{s}_f$, turning the difficulty score into a generative control knob. It closes the loop: desired bins in difficulty space translate into concrete parameter settings, ensuring principled and reproducible control over task hardness rather than reliance on uncontrolled rendering artefacts.

### B.4 BINNING STRATEGIES AND PRIORS

With a scalar difficulty score $d(\mathbf{x}) \in [0, 1]$ established, we discretise the continuum into bins that support controlled sampling during benchmark construction. Binning ensures that items are evenly distributed across difficulty levels while remaining aligned across task families.

**Quantile-based binning** We partition $d(\mathbf{x})$ into three bands: easy, medium, and hard, using global quantile thresholds. This ensures that each bin contains approximately equal probability mass, preventing trivial instances from dominating and providing adequate coverage of the hard tail. Applying thresholds globally across all task families keeps the bins comparable, so that *easy* in one class corresponds to a similar expected human effort in another. The resulting distributions across bins are shown in Figure 3a.

**Stratified priors for sampling** During synthesis, bins are sampled according to stratified priors $P_e$. These priors control the relative prevalence of easy, medium, and hard instances in the generated benchmark and are defined consistently across task families. The priors are normalised to preserve global proportions, ensuring that sampling remains balanced while still allowing targeted emphasis (e.g., for stress-testing models).

**Rejection and admissibility** After sampling from a bin, we enforce validity by rejecting any instance that violates structural constraints ($P$) or visual guards ($V$). This ensures that binning never admits ambiguous or degenerate cases, such as overlapping primitives or low-contrast distractors. The final benchmark therefore achieves a stratified and interpretable distribution of difficulty levels that is both reproducible and free of rendering artefacts.

## C SPATIAL-CAPTCHA: INVARIANT-SPECIFIED TASK MANIFESTS AND GROUND-TRUTH CERTIFICATION

### C.1 EXAMPLE MANIFEST.

To make the abstraction concrete, Listing 1 shows a minimal JSON manifest instantiating the tuple $\mathcal{M}$ for a viewpoint-matching item; each field maps to $id, I, (\Theta, P_\Theta), \mathcal{T}, \mathcal{G}, \Gamma, \mathcal{V}, \mathcal{R}$ as defined above, with the field-to-symbol alignment summarized in Table 3. The distractors are explicitly encoded as alternative agent viewpoints, validated for uniqueness, ensuring the task remains well-posed.

```
1    "type": "custom",
2    "script": "generate.py",
3    "name": "Agent Sight",
4    "input": {
5      "BOX_COUNT": {
6        "type": "int",
7        "min": 1,
8        "max": 5
```

```
 9        },
10        ...,
11        "COLOR_MAP": {
12          "type": "enum",
13          "values": ["Pastel1", "Pastel2"]
14        }
15      },
16      "task": {
17        "prompt": "Imagine you are...",
18        "answer": {
19          "num_variants": 4,
20          "variants": {
21            "type": "enum",
22            "values": ["1", "2", "3", "4"]
23          },
24          "correct": "$CORRECT"
25        }
26  }
```

Listing 1: Example JSON manifest

## D    EVALUATION PROCESS DETAILS

This appendix provides a detailed description of the evaluation metrics used throughout the study. The metrics are designed to capture not only task-level correctness but also calibration, coverage, and cognitive plausibility of model behaviour. They are computed consistently across all task families and difficulty bins.

**Pass@1** Pass@1 measures the proportion of task instances for which the model's top-ranked prediction is correct. This is the most stringent correctness metric, analogous to exact match, and reflects whether the model can reliably prioritise the correct answer without reliance on downstream ranking or sampling. Formally, if $y_i$ is the ground truth and $\hat{y}_i^{(1)}$ the top prediction for instance $i$, then

$$\text{Pass@1} = \frac{1}{N} \sum_{i=1}^{N} \mathbb{1}\{\hat{y}_i^{(1)} = y_i\}.$$

**Pass@$k$** Pass@$k$ relaxes the top-1 requirement by scoring an instance as correct if the ground truth appears within the top-$k$ predictions. This metric reflects the model's ability to maintain coverage of

Table 3: Alignment between the canonical JSON manifest and the formal tuple $\mathcal{M}$. Distractors are explicit scene variants (e.g., fake agent locations with unique views) generated alongside the correct answer.

| JSON field | $\mathcal{M}$ element | Example |
|---|---|---|
| *Metadata* | | |
| name,type,version | $id$ | `"Agent Sight","custom","1.2"` |
| *Task Semantics* | | |
| invariant | $I$ | `"view_match"` |
| task.prompt | $\mathcal{T}$ | `"Imagine you are the $TARGET in the above figure, which one of the following scenes will you see?"` |
| task.answer.correct | $\mathcal{T}$ | `"$CORRECT"` |
| *Scene & Rendering* | | |
| scene/script | $\mathcal{G}$ | `"generate.py"` |
| validators | $\mathcal{V}$ | `["uniqueness","margin"]` |
| renderer | $\mathcal{R}$ | `"custom"` |
| *Sampling & Distractors* | | |
| input.BOX_COUNT | $\Theta, P_\Theta$ | `"min":1,"max":5` |
| input.COLOR_MAP | $\Theta, P_\Theta$ | `"values":["Pastel1","Pastel2"]` |
| task.answer.variants | $\Gamma, \mathcal{G}$ | `["fake_agent_A","fake_agent_B",...]` |

Table 4: Generalized invariant families aligned with spatial–cognition abilities. Each row specifies validators that certify the intended invariant and the distractor strategies used to generate nontrivial but incorrect alternatives.

| Invariant family ($I$) | Validators ($\mathcal{V}$) | Distractor strategy ($\Gamma$) |
|---|---|---|
| Spatial perception and reference system | alignment and parallelism checks under rigid transforms; collinearity and axis consistency; uniqueness tests | near–parallel or collinear but misaligned segments; objects offset just beyond tolerance |
| Spatial orientation and perspective–taking | egocentric vs. allocentric consistency; ray–cast visibility; camera transform equivalence; uniqueness audits | fake observer viewpoints yielding plausible but incorrect views; near–pose confusions |
| Mental object rotation | rotation–equivalence under $SO(2)/SO(3)$; congruence tests with angular margins; mirror/reflection screens | mirror images; rotated near–matches differing by small angular offsets; flipped but similar silhouettes |
| Spatial visualization with multiple transformations | multi–step transformation execution (fold, revolve, unfold); graph isomorphism checks across steps; state–tracking of voxel/projection consistency | partial transformation paths; inconsistent projection sets; solids from alternative operation sequences |

the correct answer under uncertainty. For $k = 3$ as used in our study,

$$\text{Pass@}k = \frac{1}{N} \sum_{i=1}^{N} \mathbb{1}\!\!\!\not{\phantom{1}}\{y_i \in \{\hat{y}_i^{(1)}, \dots, \hat{y}_i^{(k)}\}\}.$$

$k$-**of-**$k$ **Reliability** Beyond coverage, we assess how reliably the model's top-$k$ predictions contain only correct answers. The $k$-of-$k$ metric computes the fraction of instances where *all* of the top-$k$ predictions equal the ground truth. This is stricter than Pass@$k$ and quantifies whether a high-confidence prediction set is trustworthy. For $k = 3$, this amounts to

$$\text{k-of-k} = \frac{1}{N} \sum_{i=1}^{N} \mathbb{1}\!\!\!\not{\phantom{1}}\{\hat{y}_i^{(1)} = \hat{y}_i^{(2)} = \hat{y}_i^{(3)} = y_i\}.$$

**Reliability vs. Coverage** To diagnose calibration, we plot $k$-of-$k$ against Pass@$k$ (cf. Figure 3c). Ideally, a well-calibrated model lies near the identity line: if it predicts the answer is within its top-$k$ set, then that set should be reliable. Models below the line exhibit overconfidence (claiming coverage without reliability), while those above the line are overly conservative.

**Per-ability metrics** In addition to aggregate metrics, we report Pass@1 stratified by cognitive ability class: spatial perception (SP), spatial orientation (SO), mental object rotation (MOR), and multi-step visualisation (SV). These disaggregated metrics reveal which cognitive primitives are most brittle for models and whether difficulty arises from perceptual or compositional factors.

**Human-level reference** Human annotators (N=60) provide an empirical soft upper bound. An item is considered solved if at least two annotators select the correct answer under time constraints. This yields both a pass rate and distributions of human response times, against which model predictions are normalised. Reporting both metrics provides insight into where models deviate most strongly from embodied or time-bounded human reasoning.

**Difficulty-stratified performance** Finally, we analyse metrics within Easy, Medium, and Hard bins defined by the difficulty map (Appendix B.4). This stratification verifies that accuracy decreases monotonically with difficulty for both humans and models, confirming that $d(\mathbf{x})$ captures substantive cognitive load rather than noise.

Together, these metrics provide a multi-faceted view of performance: Pass@1 captures strict correctness, Pass@$k$ captures coverage, $k$-of-$k$ exposes calibration, per-ability scores isolate cognitive bottlenecks, and human-level references provide grounding in real-world effort.

# E  FAILURE ANALYSIS

Despite modest performance on select task types, current models systematically fail to generalise spatial reasoning beyond perceptual regularities. This section analyses dominant failure modes

Table 5: Summary of evaluation metrics used in this study. Metrics are grouped by evaluation intent: correctness, calibration, efficiency, human upper bounds, and cognitive attribution.

| Metric | Type | Scope | Purpose and Interpretation |
|---|---|---|---|
| *Overall Accuracy Metrics* | | | |
| Pass@1 | Accuracy | $[0,1]$ | Top-1 correctness under deterministic decoding ($T$=0.0). Measures default model reliability without sampling. |
| Pass@k | Accuracy | $[0,1]$ | Success rate with $k$=3 completions. Probes recoverability under model uncertainty. |
| $k/k$ Reliability | Epistemic Stability | $[0,1]$ | Fraction of instances where all $k$ sampled outputs are *identical and correct*. Measures model confidence and output consistency. |
| *Human Upper Bound* | | | |
| Human-Simple Pass Rate | Sanity Check | $[0,1]$ | Fraction of instances correctly solved by at least 2 of 3 human annotators under a 30s time limit. Used to establish a baseline for "non-trick" solvability. |
| *Per-Ability Pass@1* | | | |
| Spatial Perception | Accuracy | $[0,1]$ | Accuracy on tasks requiring recognition of spatial layout, object relationships, and metric adjacency in visual scenes. |
| Spatial Orientation | Accuracy | $[0,1]$ | Accuracy on tasks involving viewpoint transformations and egocentric-to-allocentric alignment. |
| Mental Rotation | Accuracy | $[0,1]$ | Accuracy on tasks requiring rigid-body rotation of objects in 2D or 3D space. |
| Spatial Visualisation | Accuracy | $[0,1]$ | Accuracy on tasks requiring multi-step spatial transformations, such as folding, cutting, or layered movement. |

through a taxonomy of error classes and representative examples. All qualitative patterns are drawn from a held-out evaluation set, with aggregate statistics reported over n=70 Spatial CAPTCHA instances.

### E.1 TAXONOMY OF FAILURE MODES

We categorise model errors into three broad families: (i) *Invariant violations*, where the predicted output contradicts task-specified geometric or relational constraints; (ii) *Hallucinated structure*, in which the model invents non-existent elements or misattributes spatial relationships; and (iii) *Calibration errors*, wherein the top-$k$ prediction set fails to reliably include the correct answer despite high predicted likelihood.

**Invariant violations** This family dominates the error distribution, with approximately $63.5\%$ (94/148) of coded failures. In UNFOLDED, models frequently misplace facets of a cube net, breaking adjacency constraints. In PYRAMID, they misalign side-view projections, confusing planar-to-volumetric consistency. Sub-classes include *viewpoint/perspective errors* (74 cases), *rotation vs. mirror misalignments* (42 cases), and rare but diagnostic *topology/containment* violations. These patterns confirm that models fail to internalise certified invariants and instead resort to weakly correlated perceptual cues.

**Hallucinated structure** Roughly $35.1\%$ (52/148) of failures fall in this family, where models fabricate symmetry, occluded elements, or entire structures absent from the input. This is most evident in FULL VIEWS, where occluded geometry is invented, and in multi-projection tasks, where unsupported symmetries are projected onto irregular shapes. For example, GPT-4o variants tend to overgeneralise from canonical forms, inferring staircases or pyramids where no such invariants exist. These errors reveal brittle inductive priors and over-regularisation of spatial patterns.

**Calibration errors** Though less frequent in natural-language rationales, calibration issues remain evident in evaluation metrics. At $0.7\%$ of coded failures, explicit overconfidence is rare, but systematically all models show a gap between high Pass@$k$ coverage and low $k/k$ reliability. For instance, distractor options are often included in the top-$k$ set with high likelihood, while the true answer is excluded. This reflects poor uncertainty estimation and suggests that models rely on shallow scoring heuristics rather than calibrated spatial reasoning.

## F ONLINE SPATIAL CAPTCHA SERVICE

To better show our contribution on building CAPTCHA, we show the webpage screenshot of our online spatial CAPTCHA service (taking agent sight task as an example) in Figure 8.

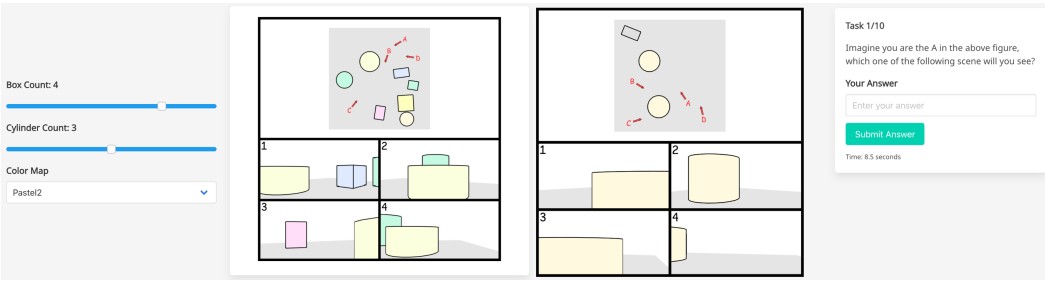

(a) Contraint-based difficulty control mechanism      (b) Question and answering page

Figure 8: Illustration of the agent sight task of our online spatial CAPTCHA service: (a) we provide difficulty control flexibility by adjusting box and cylinder counts and color maps; (b) the question and answering page which runs automated correctness verification on the backend while also recording the solving time.

## G    LLM USAGE STATEMENT

During the preparation of this manuscript, LLMs were utilized exclusively for language refinement and stylistic editing. The technical contributions, experimental design, data analysis, and interpretation of results were not generated by LLMs. All conceptual development, methodological details, coding, and evaluation are solely the responsibility of the authors. In accordance with policy, the authors assume full accountability for the accuracy and integrity of the content, and any errors or misrepresentations are exclusively their own responsibility.

## H    IMPLEMENTATION DETAILS

**Environment** All experiments were orchestrated from a local development environment running on a Mac Studio (Apple M2 Ultra, 128GB unified memory) with `Python 3.13`. However, no inference was executed locally. All model queries and evaluations were conducted via the `OpenRouter API`, ensuring a consistent inference environment across experiments.

**Human studies** For the human evaluation component, participant groups were recruited from multiple institutions and diverse demographics. In particular, we included (i) graduate and undergraduate students from two universities, and (iii) broader community participants representing varied nationalities, age groups, and professional backgrounds (recruited through the extended social networks of the authors). This composition ensured both institutional diversity and cultural heterogeneity. All human studies were conducted under informed consent protocols.

**Generation pipeline** Task generation relied on a combination of open-source 3D and visualization toolchains. Procedural scenes were synthesized using `Blender 4.4.3`, geometric manipulations and renderings were facilitated by `vedo 2025.5.4`, while classical Python libraries such as `matplotlib` were used for visualization and plotting. The generation pipeline was fully scripted and released to guarantee reproducibility.

**Validation** Automated task validation employed both standard libraries and domain-specific packages. In particular, we used `scipy==1.16.0` for statistical consistency checks and `polyomino==0.7.1` for verifying combinatorial tiling constraints. Additional validation relied on custom Python scripts to enforce task-specific invariants and to audit correctness prior to release.

**reCAPTCHA comparison** For the comparison against commercial CAPTCHA systems, we clarify that there is no publicly available reCAPTCHA-Bench. Instead, we rely on `MCA-Bench`, which contains a task type explicitly inspired by Google reCAPTCHA but manually created by the authors of MCA-Bench. To ensure fairness, we exported only the subset of MCA-Bench corresponding to reCAPTCHA-style items and used this as a proxy benchmark. This choice was motivated by the need to compare our method against the most widely deployed CAPTCHA solution in practice, while preserving as much fidelity as possible to the task format encountered in the wild.

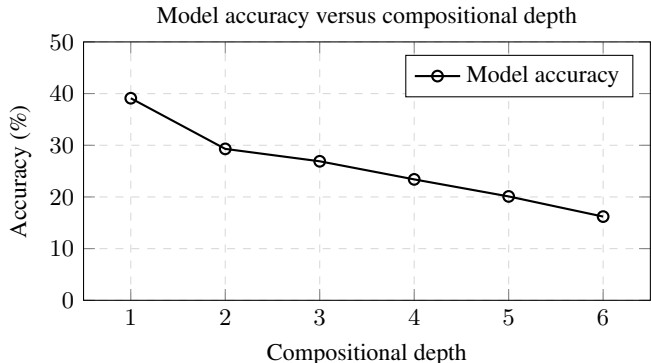

Figure 9: Model accuracy versus compositional depth in *Spatial CAPTCHA*. Accuracy declines monotonically as relational and occlusion complexity increase, confirming continuously tunable difficulty.

## I    LONGEVITY AND FORWARD-LOOKING RELEVANCE OF SPATIAL CAPTCHA

Static CAPTCHAs, once an effective mechanism for distinguishing human and machine perceptual capabilities, have exhibited rapid degradation in discriminative efficacy as vision–language and multimodal reasoning models advance. In practice, static datasets confer only transient robustness: once their distribution is absorbed into large-scale training corpora, generalisation collapses due to overfitting at the dataset or feature-template level. This phenomenon underscores a fundamental limitation: fixed perceptual benchmarks cannot sustain discriminatory power in an evolving model ecosystem.

SPATIAL CAPTCHA is designed explicitly to mitigate this brittleness through procedural generation. Rather than a static dataset, it constitutes a *parametric generative framework* that models the key dimensions of human spatial cognition. These factors jointly define a continuous control manifold from which puzzles are synthesised. By sampling along this manifold, we can systematically scale perceptual load and reasoning complexity, yielding a *continuously tunable difficulty curve*. Empirically, model accuracy degrades monotonically with increasing *compositional depth* (Figure 9), demonstrating that the generative parameters induce a smooth and controllable adjustment of challenge. This property ensures not only granular difficulty calibration but also sustained adaptability as model capabilities evolve.

### I.1    MODULAR COGNITIVE PRIMITIVES AND EXTENSIBILITY

Beyond immediate tunability, the framework is designed to remain forward-compatible with emerging insights from cognitive psychology and neuro-symbolic AI. Each cognitive primitive is implemented as a modular transformation operator within the generator. This modularity allows principled integration of new perceptual constructs without architectural reconfiguration. For example, as research elucidates mechanisms of dynamic attention shifts and anchoring in human perception (Itti & Koch, 2001) or advances in multi-object affordance modeling in robotics and embodied AI (Myers et al., 2015), corresponding operator modules can be introduced to expand the cognitive span of generated tasks.

## J    ASSESSING DATA-LIMITED VERSUS MODEL-LIMITED PERFORMANCE VIA FINE-TUNING

To determine whether the observed performance ceiling stems from data scarcity or from architectural limitations in contemporary multimodal transformers, we conduct a controlled fine-tuning study on frontier models. The objective is to assess whether additional task-specific supervision yields meaningful accuracy gains or whether the deficits instead arise from inductive biases that are misaligned with the spatial–relational structure of the benchmark.

Table 6: Spatial complexity effects where highest parameter values yield lowest accuracy. Parameters are grouped by cognitive complexity type, confirming systematic difficulty progression despite non-monotonic intermediate values.

| Parameter | Range | Accuracy (%) | Interpretation and Effect |
|---|---|---|---|
| *Baseline* | | | |
| Minimal complexity | — | 29.9 | Performance on simplest configurations (lowest parameter values). Represents baseline spatial reasoning capability before complexity perturbations. |
| *Visual Complexity* | | | |
| Scene density (2–10 objects) | Low–High | 30.3 | Accuracy drops from 42.2% (sparse) to 25.8% (dense scenes). Visual clutter and occlusion create systematic processing challenges despite non-monotonic intermediate values. |
| *Geometric Complexity* | | | |
| Polygon complexity (3–6 sides) | Low–High | 28.0 | Performance decreases from 29.0% (triangles) to 21.7% (hexagons). Higher-order polygons increase geometric reasoning demands despite intermediate fluctuations. |
| *Combinatorial Complexity* | | | |
| Spatial arrangement (3–5 tiles) | Low–High | 25.8 | Performance declines from 27.8% to 23.2% with more pieces. Increased combinatorial demands strain spatial working memory and arrangement reasoning. |
| *Projection Complexity* | | | |
| Mapping resolution (3×3–4×4) | Low–High | 24.7 | Accuracy decreases from 26.6% to 22.7% with finer grids. Higher resolution increases 3D-to-2D correspondence complexity. |
| *Surface Complexity* | | | |
| 3D surface detail (4–8 vertices) | Low–High | 21.4 | Accuracy drops from 22.7% to 19.7% at highest vertex counts. Complex surfaces challenge 3D transformation understanding despite non-linear progression. |

## J.1 FINE-TUNING CONFIGURATION

We fine-tuned the most capable publicly accessible frontier model at the time of study (gpt-4o (2024-08-06 base)) via the official OpenAI supervised fine-tuning interface. Table 7 summarises the exact hyperparameters and training budget.

Table 7: Fine-tuning hyperparameters and training budget.

| Parameter | Value |
|---|---|
| *Model and Data Configuration* | |
| Model | gpt-4o (2024-08-06 base) |
| Training tokens | $\approx$ 3.7M |
| Epochs | 3 |
| Batch size | 2 |
| *Training Dynamics* | |
| Learning-rate multiplier | 2× |
| Optimisation mode | Supervised fine-tuning (OpenAI API) |
| Reasoning budget | Fixed; identical to evaluation setting |
| Prompt scaffold | Unchanged; only task-specific examples added |

To reduce confounds related to prompt-format drift or context-length truncation, we retained the original evaluation scaffold, modifying only the demonstrations used during fine-tuning.

## J.2 EFFECT OF FINE-TUNING ON ACCURACY

Fine-tuning produced a modest improvement: accuracy rose from 38% to 57.7%, still far below human performance (89.5%). Table 8 provides the full comparison, including the stronger gpt-5-1 model.

Despite increased model scale and reasoning capacity, gpt-5.1 exhibits only marginal gains, mirroring the stagnation observed after fine-tuning.

## J.3 IMPLICATIONS

Taken together, the evidence supports a model-limited interpretation: current multimodal transformers appear constrained not by exposure to the task distribution but by representational inadequacies

Table 8: Accuracy improvements from fine-tuning, compared against stronger baseline models and human performance.

| System | Accuracy (%) |
|---:|:---:|
| *Model Performance* | |
| gpt-4o (base) | 38.0 |
| gpt-4o-2024-08-06 (fine-tuned; ours) | 57.7 |
| *Comparative Upper Bounds* | |
| gpt-5.1 | 61.5 |
| Human | 89.5 |

intrinsic to their architecture. These include weak mechanisms for modelling equivariance, non-local geometric dependencies, and viewpoint-consistent relational structure. Mitigating such limitations likely requires architectural interventions (e.g., explicit spatial modules, equivariant layers, or hybrid neural–symbolic operators) rather than additional data alone.

### REPRODUCIBILITY NOTES

All fine-tuning runs adhered strictly to the standard OpenAI API configuration, without undocumented hyperparameter overrides. Logs, prompts, and training traces appear in the supplementary artefacts.

## K PERCEPTUAL CLARITY SAFEGUARDS

To ensure that logically well-formed SPATIALCAPTCHA instances also exhibit robust *perceptual* unambiguity, we implement a three-stage verification pipeline: (i) *pre-deployment human validation*, (ii) *post-render robustness checks*, and (iii) *runtime quality control*.

### K.1 HUMAN PERCEPTUAL VALIDATION

A dedicated perceptual-clarity study demonstrates that more than *97 %* of generated puzzles achieve at least *95 % inter-participant agreement*. The empirical distribution confirms that visually ambiguous instances constitute only a negligible minority, providing a lower bound on perceptual soundness prior to any algorithmic filtering.

### K.2 AUTOMATED POST-RENDER VERIFICATION

Puzzle families subject to strong 3D/2D projection artefacts (e.g., viewpoint rotation or perspective transformations undergo an automated similarity-based confusability check). Rendered silhouettes of the target and distractors are compared; instances exceeding a calibrated similarity threshold are discarded.

### K.3 RUNTIME RELIABILITY-WEIGHTED FILTERING

During deployment, SPATIALCAPTCHA continuously monitors online human-response consistency. Puzzles exhibiting anomalously high disagreement (relative to historical baselines for their category) are automatically flagged and removed. This procedure provides adaptive quality control under distributional drift and long-tail ambiguity, mirroring mechanisms used in large-scale production CAPTCHA systems.

Together, these safeguards ensure that logical correctness is matched by high perceptual clarity, enforced both at generation time and under live conditions.

## L    POLICY-INDUCED ASYMMETRIES IN CAPTCHA PERFORMANCE

The apparent advantage of open-source models on the Spatial-CAPTCHA benchmark is predominantly a *policy-induced abstention artefact* rather than evidence of superior visuospatial reasoning. Contemporary proprietary MLLMs incorporate multi-layered safety and anti-abuse filters that aggressively pattern-match canonical reCAPTCHA affordances (e.g., characteristic logos, tiled layouts, and lexical cues such as "captcha", "are you a robot?", or embedded site-key prompts). These filters frequently trigger deterministic refusal templates (e.g., "I cannot assist with CAPTCHAs"), thereby converting a large fraction of queries into hard abstentions.

By contrast, our Spatial-CAPTCHA are intentionally unbranded and semantically neutral. They lack the visual and textual markers that typically activate CAPTCHA-protection policies, and are therefore interpreted by proprietary systems as generic geometric-reasoning tasks. Open-source models, which usually ship without hard refusal rules for CAPTCHA-adjacent content, attempt both tasks uniformly. As a result, their measured accuracy reflects a greater proportion of unblocked attempts rather than a genuine cognitive advantage.

**Refusal-Aware Evaluation.**    To disambiguate reasoning capacity from policy-induced abstention, we report for each model and task (i) the proportion of explicit refusals (REFUSAL%) and (ii) overall accuracy across all attempts (ACCURACY%). This refusal-aware stratification (see Table 9) enables fairer cross-model comparisons and reveals that the primary driver of observed performance asymmetry is the refusal layer rather than underlying model competence.

Table 9: Model performance on reCAPTCHA tasks with refusal-aware reporting.

| Model | Refusal (%) | Accuracy (%) |
|---:|:---:|:---:|
| claude-sonnet-4 | 10.0 | 11.3 |
| claude-opus-4 | 6.7 | 10.0 |
| qwen2.5-vl-72b-instruct | 0.0 | 2.7 |
| llama-4-maverick | 0.0 | 1.3 |

