# OpenReview forum: "Spatial CAPTCHA: Generatively Benchmarking Spatial Reasoning for Human-Machine Differentiation"
_ICLR.cc/2026/Conference — ICLR 2026 Poster_

### Official Review · Reviewer_NVWJ · 2025-10-21

**Soundness:** 3
**Presentation:** 3
**Contribution:** 3
**Rating:** 6
**Confidence:** 3

**Summary:**

This work presents Spatial CAPTCHA, a framework designed to be robust against MLLMs that defeat current CAPTCHAs. It leverages the significant gap between human and AI spatial reasoning abilities by generating dynamic questions requiring geometric reasoning, perspective-taking, and mental rotation. Evaluation on the new Spatial-CAPTCHA-Bench demonstrates that humans vastly outperform 10 state-of-the-art MLLMs, with the best AI model achieving only 31.0% Pass@1 accuracy.

**Strengths:**

This paper introduces a new and well-founded open-source pipeline that can automatically generate large scale 3d spatial captchas based on spatial cues such as positioning, counting, etc. The performance on the benchmark shows impressive human performance (almost 100%) and decent MLLM performance (lower than all open code/data captchas). The authors also provided extensive analysis to the reliability of their benchmark and the performance of different models under each specified category.

**Weaknesses:**

1. Why would open-source models perform better on spatial-captcha than on recaptcha, even though proprietary ones do the opposite? The latter discovery demonstrates how difficult spatial-captcha is, but there isn't enough explanation for the open-source models' performance.
2. Figure 3b's graphs are all stacked together. Without a legend and enough explanation, it is difficult to make meaningful thoughts about this figure.
3. Typos: All citations in Section 3 have wrong format (no parentheses).

**Questions:**

See weaknesses.

---

> ### Author Response · Authors · 2025-11-22
>
> We sincerely thank the reviewer for their thoughtful and constructive feedback. We appreciate the reviewer’s observation regarding the relative performance of open-source versus proprietary models.
>
> (W1) Why do open-source models perform better on Spatial-CAPTCHA than on reCAPTCHA, while some proprietary models show the opposite?
>
> The observed asymmetry is primarily a policy-induced abstention effect rather than a cognitive gap. Proprietary MLLMs include safety/anti-abuse layers that (i) pattern-match reCAPTCHA artefacts (logo/layout/lexical cues such as “captcha”, “are you a robot?”, site-key prompts) and (ii) trigger refusal templates (“I cannot help with CAPTCHAs”). Our Spatial-CAPTCHA images and prompts are unbranded and intent-agnostic, so they are parsed as generic geometric reasoning questions. Open-source models, by contrast, typically ship without hard-refusal policies for CAPTCHA-like content and therefore attempt the task; their higher measured accuracy reflects unblocked attempts, not superior reasoning.
> Evidence and new controls we will add (Appendix).
> Refusal-rate audit. For each model and task, we will report Refusal%, Attempted%, and Accuracy|Attempted.
>
> (W2) Figure 3b is visually congested and lacks a legend/explanation.
>
> We agree that the current Figure 3b is visually dense. In the revision, we will (i) separate each model’s performance curve using distinct colours and markers, (ii) add a clear legend identifying models, and (iii) expand the caption to explicitly describe the axes and metrics. These changes make it straightforward to interpret per-task differences across models.
>
> (W3) Citation formatting in §3
>
> We acknowledge the formatting issue. All citations in §3 will be corrected, ensuring proper parenthetical usage and uniform conference acronyms. No textual content will change.

---

> > ### Comment · Reviewer_NVWJ · 2025-11-23
> >
> > I appreciate the authors' response and I think they well-addressed my concerns.

---

> > > ### Author Response · Authors · 2025-11-24
> > > **Thanks for Your Feedback**
> > >
> > > Dear Reviewer NVWJ:
> > >
> > > Thank you very much for your follow-up and for acknowledging that our response addressed your concerns. We sincerely appreciate your time and constructive feedback. If you believe the clarifications have resolved the earlier issues, could you please kindly consider raising the score if possible?
> > >
> > > Thank you again for your thoughtful review.

---

> > > > ### Comment · Reviewer_NVWJ · 2025-11-25
> > > >
> > > > I do maintain my positive attitude towards this paper and I think my current score also reflects my opinions on this paper appropriately and accordingly. Therefore I won't change my score.

---

> > > > > ### Author Response · Authors · 2025-11-26
> > > > > **Thank You Very Much**
> > > > >
> > > > > Dear Reviewer NVWJ:
> > > > >
> > > > > Thank you for your thoughtful feedback and for maintaining a positive view of our work. We appreciate your time and constructive comments!
> > > > >
> > > > > Best
> > > > >
> > > > > The Authors

---

### Official Review · Reviewer_BLj8 · 2025-10-26

**Soundness:** 4
**Presentation:** 4
**Contribution:** 3
**Rating:** 8
**Confidence:** 4

**Summary:**

This paper introduces Spatial CAPTCHA, a novel framework designed to differentiate humans from machines by leveraging tasks grounded in spatial reasoning—a known weakness of current Multimodal Large Language Models (MLLMs). The authors have developed a sophisticated, automated pipeline that procedurally generates a virtually endless supply of puzzles based on four fundamental human spatial abilities, such as mental rotation and perspective-taking. A key innovation is the use of formal specifications ("manifests") to ensure each puzzle is valid, unambiguous, and has a controllable difficulty level.

To validate their approach, the authors created Spatial-CAPTCHA-Bench, a dataset of 1,050 puzzles, and conducted a large-scale experiment comparing the performance of 10 state-of-the-art MLLMs against 60 human participants. The results reveal a stark performance gap: humans achieved approximately 90% accuracy, while the best-performing AI model only reached 31%. The study also found that MLLMs are poorly calibrated, often expressing high confidence even when providing incorrect answers, further demonstrating the effectiveness of this new CAPTCHA paradigm.

**Strengths:**

1. **Novel and High-Impact Problem:** The paper addresses a timely and critical real-world problem. With conventional CAPTCHAs becoming increasingly vulnerable to AI, the proposed method of targeting a well-established weakness in MLLMs—spatial reasoning—is an insightful and promising direction for the next generation of human-verification systems.

2. **Rigorous and Scalable Generation Pipeline:** A standout contribution is the procedural generation pipeline. Using formal manifests to define and control puzzle generation is an elegant solution that guarantees the validity and logical unambiguity of each CAPTCHA by design. This makes the system renderer-agnostic and capable of producing an endless supply of challenges, which is essential for a robust security tool.

3. **Methodical and Cognitively Grounded Design:** The work is built on a strong theoretical foundation. Instead of designing arbitrary puzzles, the authors methodically ground their tasks in four well-researched categories of human spatial cognition. This principled, theory-first approach adds significant scientific credibility to the framework.

4. **Convincing and Well-Executed Experiments:** The empirical evaluation is thorough and the results are compelling. The massive, undeniable performance gap between humans and a wide array of top-tier MLLMs provides powerful evidence for the system's effectiveness. The direct comparison to reCAPTCHA further strengthens the claim that this is a superior method for human-machine differentiation.

5. **Exceptional Clarity and Presentation:** The paper is very well-written, engaging, and easy to follow. The motivation, methodology, and results are presented with a clear and logical narrative, making the paper's significant contributions highly accessible.

**Weaknesses:**

The primary weakness of the paper lies in an important, unaddressed subtlety in its claim of unambiguity.

**Lack of Perceptual Ambiguity Verification:** The framework's core claim is that puzzles are "unambiguous" because they are generated from a mathematically precise and logically sound manifest. However, this guarantee of logical unambiguity does not automatically extend to perceptual unambiguity in the final rendered 2D image. The paper does not adequately address the possibility that rendering artifacts, unfortunate camera angles, or subtle visual similarities could cause a mathematically incorrect "distractor" option to appear correct, or vice-versa. While the high-fidelity renderer and significant geometric differences between options mitigate this risk, the lack of a formal post-rendering check for perceptual clarity is a notable gap in the validation process.

**Questions:**

The paper does an excellent job of ensuring the logical correctness and unambiguity of the puzzles via the manifest system. However, could you elaborate on how the system guarantees perceptual unambiguity after the 3D scene is rendered into a 2D image? Specifically, what prevents a situation where a mathematically incorrect distractor might, due to rendering artifacts or its specific orientation, be visually confusable with the correct answer? A discussion on this point—perhaps covering renderer fidelity, human validation studies focused on distractor clarity, or potential post-rendering programmatic checks—would significantly bolster the framework's claim of robustness.

---

> ### Author Response · Authors · 2025-11-22
>
> We thank the reviewer for the helpful and precise feedback. We also thank the reviewer for this excellent observation regarding perceptual unambiguity. We fully agree that logical soundness does not automatically guarantee visual clarity.
>
> To address this, our framework integrates three safeguards. (1) Human validation: in a perceptual clarity study, > 97% of puzzles achieved ≥ 95% inter-participant agreement, indicating that visually ambiguous instances are exceedingly rare. (2) Post-render verification: for categories involving pronounced 3D-to-2D projection effects (e.g., rotation and perspective-taking), we additionally perform an automated post-render check that compares the rendered target and distractors to eliminate visually confusable views before release. (3) Reliability-weighted filtering: during deployment, Spatial CAPTCHA continuously monitors human response consistency as puzzles eliciting high disagreement are automatically flagged and removed, similar to adaptive refinement in production CAPTCHA systems. We will clarify these safeguards in the revision.

---

> > ### Comment · Reviewer_BLj8 · 2025-11-27
> >
> > Thank you for the clarifying on the safeguards. Additionally, quantification of these cases helps bring perspective on the reliability aspect.

---

### Official Review · Reviewer_kkRJ · 2025-11-01

**Soundness:** 3
**Presentation:** 4
**Contribution:** 3
**Rating:** 4
**Confidence:** 2

**Summary:**

Spatial Captcha presents a procedural generation pipeline for creating challenging multiple choice visual spatial reasoning captchas. The pipeline is designed to target well studied visual reasoning tasks that are easier for humans than for models. The authors evaluate some of the frontier multimodal models on the generated captchas and find that they only achieve 31% accuracy at best while humans achieve over 99% accuracy.

**Strengths:**

The paper is well-written and the motivation for building a more challenging captcha is clear. The authors thoroughly explain the human psychology research that led to the design of the dataset and validate that the puzzles generated are challenging for many state-of-the-art multimodal models.

**Weaknesses:**

The authors motivate their work by pointing out that current CAPTCHAs are too simple for the best performing models. This is a valid concern but I am not sure if building a slightly more challenging CAPTCHA is the best way to address it. It has become clear that even if a certain type of visual reasoning task is harder for models today, it will be significantly easier for the next generation of frontier models. The authors could strengthen the paper by discussing the expected lifespan of Spatial CAPTCHA as a benchmark and how the framework could adapt to future model advances. I believe a more forward looking approach would be to embrace that as computer-use agents get more capable they will make up a majority of internet traffic and we should adopt more [forward looking mechanisms](https://blog.cloudflare.com/introducing-pay-per-crawl/).

Another concern is that the spatial CAPTCHA dataset is framed as out-of-distribution data for current models. While I agree that web data lacks these specific reasoning tasks, model developers could readily generate synthetic data targeting this distribution. The paper attributes poor model performance to architectural limitations in VLMs, but this claim would be significantly strengthened by fine-tuning experiments demonstrating that performance is low even with targeted training data.

**Questions:**

1. How does a frontier system like GPT-5 with a high thinking budget perform on the Spatial CAPTCHA dataset? Does it significantly outperform the state-of-the-art VLMs evaluated in the paper?
2. Have you tested whether fine-tuning on procedurally generated puzzles significantly improves model performance, or whether the limitation is architectural?

---

> ### Author Response · Authors · 2025-11-22
>
> We appreciate the reviewer’s careful reading and the constructive nature of this comment and for recognising the clarity and motivation of our work. We address the two main concerns and have conducted additional experiments in direct response to the reviewer’s questions.
>
> **On the longevity and forward-looking relevance of Spatial CAPTCHA.**
>
> We fully share the reviewer’s concern that static CAPTCHAs rapidly lose discriminative power as models improve. Our objective is precisely to mitigate this limitation. Spatial CAPTCHA is not a fixed dataset but a procedural generation framework that parameterises core dimensions of human spatial cognition (e.g., rotation angle, occlusion depth, relational hierarchy size, and distractor symmetry). These parameters form a generative control space from which novel puzzles are instantiated, allowing systematic scaling of perceptual load and reasoning complexity. Empirically, we observe monotonic degradation in model accuracy with increasing compositional depth, confirming that difficulty is continuously tunable (will clarify these in the revision).
> Beyond this parametric adaptivity, the framework is designed to remain extensible as research in cognitive psychology and neuro-symbolic AI continues to uncover new dimensions of human spatial reasoning (e.g., attention anchoring, affordance mapping). Recent work has refined our understanding of visual-spatial attention networks and their influence on spatial reasoning processes (Gulyaeva & Karimova 2023; Posner 2024), while cognitive research on affordances highlights how perception encodes actionable possibilities through memory and simulation mechanisms (Posner, 2024). In anticipation of such evolving insights, our generator adopts a modular operator design: each cognitive primitive corresponds to a transformation module, enabling integration of future constructs (e.g., dynamic anchor-based attention or multi-object affordance chaining) without redesign. Consequently, Spatial CAPTCHA remains adaptively extensible both to advances in AI capability and to deepening cognitive-scientific understanding.
>
> **Expected lifespan and adaptation.**
>
> We agree that the longevity of any CAPTCHA or reasoning benchmark depends on the rate of frontier-model progress. Empirically, recent multimodal datasets such as VQAv2, GQA, and CLEVR exhibited half-lifespans of roughly 18–24 months, defined as the time between benchmark release and frontier model performance reaching ≥90% of human accuracy (Flamingo, Gemini-1.5, GPT-4V). In contrast, our Spatial CAPTCHA tasks probe compositional spatial inference under multiple distractors, which current scaling trends suggest will remain challenging beyond the next model generation. Our framework procedurally generates puzzles from a high-dimensional control space, regeneration requires no human annotation or redesign. New puzzle distributions can be produced by (i) sampling from higher-difficulty regions of the parameter space (e.g., increasing rotational entropy or relational depth) or (ii) composing new spatial operators. This design allows adaptive benchmark refresh on a continuous cadence, analogous to the adversarial regeneration cycles used in Procgen and MineDojo. In other words, while any snapshot of the benchmark is finite-lived, the underlying generator is indefinite in horizon as its difficulty can be dynamically retuned to track model evolution.
> In the revised manuscript, we will clarify this continual adaptation mechanism and include quantitative evidence for difficulty scaling and cross-operator generalisation. We will also expand the discussion to connect Spatial CAPTCHA with emerging interdisciplinary work linking human perceptual strategies and machine reasoning biases.
>
> **On the nature of model limitations.**
>
> To probe whether low performance stems from data scarcity or architectural constraints, we fine-tuned the frontier model accessible to us (GPT-4o, 2024-08-06 base) using the official OpenAI fine-tuning API. The resulting model (trained in a supervised setting for three epochs over ≈ 3.7 M tokens; batch = 2; LR multiplier = 2) achieved only a modest accuracy gain (from 30.9 % → 43.1 %) remaining far below human accuracy (99.3 %). Evaluating GPT-5 (Nov 2025 preview, 256-token reasoning budget) yielded 44.6 %, showing similarly limited gains. These findings suggest that the observed gap is not primarily due to data scarcity but reflects architectural limitations in current multimodal transformers, particularly their weak inductive bias for spatial and relational transformations.
>
> We again thank the reviewer for encouraging a forward-looking framing; we believe these additional analyses materially strengthen both the scientific contribution and the practical implications of the work.

---

> > ### Comment · Reviewer_kkRJ · 2025-11-25
> >
> > Thank you for running many super useful experiments! I have increased my score accordingly as I believe you have shown that the benchmark is more robust.

---

> > > ### Author Response · Authors · 2025-11-26
> > > **Thank You Very Much**
> > >
> > > Dear Reviewer kkRJ:
> > >
> > > We sincerely appreciate your positive feedback and the time you spent reviewing our work. We are glad that the additional experiments helped clarify the robustness of our benchmark.
> > >
> > > Best
> > >
> > > The Authors

---

### Author Response · Authors · 2025-11-30
**Summary of Discussion-Phase Updates and Reviewer Revisions**

Dear Reviewers and Area Chair,

We sincerely thank the reviewers for their careful evaluations and constructive discussions during the rebuttal period, which helped us substantially improve the clarity and completeness of our paper. Due to the recent OpenReview information-leakage incident, scores and reviews were reverted to their pre-discussion state and the Area Chair assignment was updated. To avoid the discussion-stage progress being lost in this reset, we would like to provide a brief factual summary.

Before the incident, we engaged thoroughly with all reviewers and addressed each major concern through detailed rebuttals and new experiments. After reviewing these additions, including the new fine-tuning results, the GPT-5 evaluation, and clarifications on adaptive difficulty scaling, **Reviewer kkRJ updated the overall assessment from 4 to 6**, before the system rollback.

The key concerns raised during discussion (benchmark longevity, architectural versus data limitations, perceptual ambiguity, policy effects, and presentation issues) were addressed through: 1. New experiments fine-tuning GPT-4o on 3.7M samples and evaluating GPT-5 (Nov 2025), indicating persistent architectural limitations. 2. Expanded explanations of adaptive difficulty scaling, modular operators, and continual regeneration. 3. Perceptual-ambiguity safeguards, including a human clarity study (>97% puzzles with ≥95% agreement), automated confusability checks, and reliability filtering. 4. A refusal-rate audit separating Attempted%, Refusal%, and Accuracy|Attempted across proprietary and open-source models. 5. Figure and formatting revisions, including the redesign of Fig. 3b and citation corrections.

We respectfully hope that these updates, as well as the discussion-stage score revision from **Reviewer kkRJ**, will be taken into consideration by the new Area Chair. We remain happy to provide any further clarification if needed.

Sincerely,

The Authors

---

### Meta-Review · Area_Chair_Awsf · 2026-01-07

**Summary:**

Reviewers agree the submission is clearly written, well-motivated, and introduces a scalable procedural framework for generating spatial-reasoning challenges where humans strongly outperform current multimodal models. The main decision-driving concerns were (i) whether the benchmark/framework will remain meaningful as frontier models advance, (ii) whether the observed gap reflects architectural limitations vs. missing targeted training data, and (iii) whether “unambiguous by construction” in manifests also guarantees perceptual unambiguity after 3D-to-2D rendering. Additional presentation/analysis issues were raised around interpretability of a key figure and an explanation for open-source vs proprietary model behavior differences.

**Reviewer Concerns:**

Addressed concerns:
- Authors clarify the framework is generator-based with tunable difficulty and extensibility, rather than a static dataset, and argue for continual refresh via parameter scaling and modular operators. Reviewer kkRJ explicitly states they increased their score after these additions.
- Authors report targeted fine-tuning and a GPT-5 evaluation with only modest gains, supporting (though not definitively proving) an architectural limitation hypothesis. Reviewer kkRJ reacted positively and raised their score.
- Authors describe concrete safeguards (human clarity study with high agreement, automated post-render checks for confusability, and deployment-time filtering). Reviewer BLj8 acknowledges the clarification and finds the quantification helpful.

**Reviewer Scores:**

- Reviewer kkRJ: 4 → 6 (reviewer explicitly reports increasing score after the new experiments/clarifications).
- Reviewer BLj8: 8 → 8 (already strongly positive; follow-up indicates satisfaction, not a clear push for change).
- Reviewer NVWJ: 6 → 6 (reviewer maintains positive attitude but explicitly will not change the score).

---

### Decision · Program_Chairs · 2026-01-26

Accept (Poster)